# The relationship between postpartum depression and appropriate infant feeding practice in eastern zone of Tigray, Ethiopia: A comparative cross-sectional study

Angesom Weldu[1]*, Ayele Belachew[2], Mengistu Yilma[2]

1 Department of Epidemiology, School of Public Health, Mizan Tepi University, Tepi, Ethiopia, 2 Department of Preventive Medicine, School of Public Health, Addis Ababa University, Addis Ababa, Ethiopia

* angewell123@gmail.com

**Data Availability Statement:** All relevant data are within the paper and its Supporting Information files.

## Abstract

### Background

Understanding the relationship between postpartum depression and infant feeding practice may help to reduce the indirect impact of postpartum depression on infant feeding practice. This will further have a positive impact on reducing infant morbidity and mortality attributed to improper feeding practices. Although studies in the country have assessed the prevalence of infant feeding practices, those assessing the association between postpartum depression and infant feeding practices are lacking. Therefore, this study aimed to compare appropriate infant feeding practices and their associated factors among postpartum depressed and non-depressed mothers in Eastern Tigray.

### Methods

A comparative cross-sectional study was conducted from March 2019 to April 2019. A multi-stage random sampling technique was used to select 171 mothers with postpartum depression and 342 mothers without postpartum depression. Data were collected using a structured questionnaire from the Monitoring and Evaluating for Breastfeeding Practices toolkit, then entered into Epi- info and exported into SPSS for further analysis. A binary logistic regression was applied to determine the association between postpartum depression and appropriate infant feeding practice.

### Results

The overall prevalence of appropriate infant feeding practice was 37.6% (95% CI: 33.5%-41.9%). The prevalence was higher among mothers without postpartum depression 42.7% (95% CI: 42.9%-53.2%) than among postpartum depressed mothers 27.5% (95% CI: 24.7%-32.5%). The odds of appropriate infant feeding practice among mothers with infant birth orders of three or above was 58% (AOR = 0.42; 95% CI: 0.26–0.97) less than those mothers with infant birth orders of three and below. Households with monthly income 1000–1999 ETB (AOR = 2.26; 95% CI: 1.01–5.08), 2000–2999 ETB (AOR = 1.96; 95% CI: 1.21–

**Funding:** The source of funding for this study was from Addis Ababa University and the funders had no role in study design, data collection and analysis, decision to publish, or preparation of the manuscript.

**Competing interests:** The authors have declared that no competing interests exist.

**Abbreviations:** ANC, Antenatal Care; AOR, Adjusted Odds Ratio; CI, Confidence Interval; COR, Crude Odds Ratio; EDHS, Ethiopian Demographic and Health Survey; ETB, Ethiopian Birr; EBF, Exclusive Breast Feeding; IYCF, Infant and Young Child Feeding; KA-HDSS, Kilte Awlaelo Health and Demographic Surveillance Site; PPD, Postpartum Depression; PNC, Postnatal Care; SPSS, Statistical Package for Social Science; SRQ-20, Self-Reporting questionnaire-20; USAID, United States Agency for International Development; WHO, World Health Organization.

4.73) and 3000–3999 ETB (AOR = 5.13; 95% CI: 1.97–13.4) were more likely to practice appropriate infant feeding.

## Conclusion

The overall prevalence of appropriate infant feeding practices in the study area was low. A significantly higher proportion of mothers without postpartum depression practice appropriate infant feeding compared to mothers with postpartum depression. In addition, households with higher monthly incomes and mothers with infant birth orders three or above were significant determinants of appropriate infant feeding practice. Therefore, strengthening the provisions of nutritional education, integrating maternal mental health with routine maternal health care services, providing economic support to mothers with low income, and health education for multiparous women is a critical interventions to improve appropriate infant feeding practice.

## Introduction

World Health Organization (WHO) stated appropriate infant feeding practice for all infants who are exclusively breastfed for the first six months and receive nutritionally adequate and safe complementary foods while breastfeeding continues for up to two years of age or beyond.

Poor nutrition results from a lack of food and inappropriate feeding practices where the timing, quality, and quantity of foods given to infants are often inadequate and unsafe. Appropriate feeding practice reduces many risks, which contribute to persistent child malnutrition, mortality and morbidity, and further helps improve the nutritional status of children under two years of age [1].

Globally, inappropriate infant feeding practices are attributed to over two-thirds of these 10.9 million deaths, and suboptimal infant and young child feeding practices are attributed to 45% of all child deaths in which two-thirds of those deaths occurred in the first year of life [2]. Optimal breastfeeding could prevent 13% of deaths in children under the age of five, while appropriate complementary feeding practices could result in an additional 6% reduction in under-five mortality [3].

Although evidence of the life-saving benefits of appropriate infant feeding practices is compelling, only 58% of 0–5 month infants were exclusively breastfed in Ethiopia and 11% of infants began complementary foods before 6 months of age, which is contrary to the WHO recommendation [4].

Postpartum depression is a nonpsychotic mood disorder affecting 10–15% of women in the developed country [5], and 19.8% of women in low- and middle-income countries [6]. This proportion would be higher if the tools to measure postpartum depression were contextualized culturally and used the same cut-off points for diagnosis [7].

Unmanaged postpartum depression has negative consequences for both mothers and infants. The maternal consequences include lower quality of life, impaired social function, and poor psychological health. Increasing the risk of perceived stigma in the community could lead mothers with postpartum depression to be left undiagnosed, which causes a vicious cycle [8, 9]. The combined effect of these elements compromises maternal caregiving quality and creates an unconducive environment for optimal infant feeding practice.

Different studies have shown that maternal common mental disorders increase the risk of inappropriate infant feeding practices [10–12]. According to a study conducted in Australia, maternal psychosocial factors have a greater potential influence on exclusive breastfeeding than known less modifiable socio-demographic factors [13].

Despite the high prevalence of postpartum depression and malnutrition in the eastern zone of Tigray, Ethiopia [14, 15], studies that assess the association between postpartum depression and infant feeding practice are lacking. This study has important implications for the early identification of women who are at risk for developing postpartum depression and the implementation of screening strategies for postpartum depression. Therefore, this study aimed to examine the association between postpartum depression and appropriate infant feeding practices.

## Materials and methods

### Study area and design

A community-based comparative cross-sectional study was conducted in Kilte Awlaelo-Health and Demographic Surveillance Site (KA-HDSS), Eastern zone of Tigray, Ethiopia from March to April 2019. KA-DHSS has 9 rural and 3 urban kebeles (the smallest administrative units) and is located about 835 kilometers (KMs) north of Addis Ababa, the capital city of Ethiopia. The total population and households in KA-DHSS are 104,464 and 21,485, respectively, and are characterized by agro-climatic conditions and predominantly rural inhabitants. In the study area, under-five children and infants accounted for 8492 and 2247, respectively. There are five health centers, ten health posts, and one hospital which provide health services to the community.

### Study participants

The source population were all mothers of children aged less than 12 months living in the eastern zone of Tigray. All mothers of children aged less than 12 months in the selected households during the data collection period were the study population.

### Eligibility criteria

All mothers of children aged less than 12 months and mothers who had lived for at least six months in the study area were included in this study. Those mothers who are below the age of 18 years old and unable to communicate were excluded from the study.

### Sample size determination

The minimum sample size was determined using the double population proportion formula with the following assumptions: the proportion of unsuccessful breastfeeding among non-postpartum depressed mothers (considered as unexposed) was estimated to be 49.8% [16]. Then, the sample size required for detecting an odds ratio of 2, at 95% confidence interval, power of the study 80%, and unexposed to exposed ratio of 2:1. Based on the above assumptions, the minimum calculated sample size was 167 exposed mothers, and 334 unexposed mothers were required. By applying a design effect of 1.5 and 5% non-response rate, a total of 175 exposed mothers, with postpartum depression and 350 unexposed mothers, without postpartum depression were required.

### Sampling technique and procedure

A multistage simple random sampling was used to recruit study participants. KA-HDSS has 12 kebeles. Five of these kebeles were selected randomly using a lottery method, which included

three rural and two urban kebeles. Using a coding system and the assistance of health extension workers, we identified the households of eligible mothers in selected kebeles. As a result, we identified 1236 households in selected kebeles with mothers of children aged less than 12 months.

We conducted a census in the selected kebeles and then identified mothers with and without postpartum depression through face-to-face interviews of SRQ-20. The total size of study participants was proportionally allocated to the total infant size in each selected kebele in KA-HDSS to ensure the representativeness of the study participants. For each infant of a depressed mother, two infants of a non-depressed mother were selected from the same kebeles. For those mothers who had more than one infant at study time, the last-born infant was selected.

## Data collection tools and procedure

Infant feeding practices were assessed using a questionnaire adapted from a tool kit for monitoring and evaluating breastfeeding practices and programs, and the WHO's indicator guidelines for assessing infant and young child feeding practices [17, 18]. Infant feeding practices were assessed based on a 24-hr recall method, which reflects the feeding practices of an infant one day before the interview. In this study, appropriate infant feeding practice was assessed based on adherence to WHO recommended practices. According to WHO recommended practice, only exclusive breastfeeding was considered appropriate for infants aged 0–5 months, so infants who received prelacteal foods, liquids, and solids other than vitamins and medicine were considered inappropriate. Complementary feeding practice was assessed using indicators recommended by WHO for infants aged 6–11 months, which include timely initiation of complementary foods at six months, minimum dietary diversity, minimum meal frequency, and minimum acceptable diet. Complementary feeding practice was considered appropriate if all four indicators mentioned above were fulfilled, otherwise, it was considered inappropriate.

Postpartum depression was assessed using the self-reporting questionnaire-20(SRQ-20) developed by WHO to screen for psychiatric disturbances in developing countries. The instrument contains 20 items that ask about depressive, anxiety, and somatic symptoms present in the preceding 4 weeks. The possible responses are 'NO' coded as 0 and 'YES' as 1, with a minimum and maximum score of 0 and 20, respectively. Mothers who had SRQ-20 scores of six or above were identified as having postpartum depression. A cut-off score of $\geq 6$ yes to SRQ-20 questions was taken as it has been evidenced to have excellent internal consistency (Cronbach's alpha = 0.84) and receiver operator characteristic (ROC) curve analysis showed an area under the curve of 0.87 [19, 20]. The questionnaire mainly addressed socio-demographic, infant feeding practice, postpartum depression, and maternal health care related characteristics.

## Study variables

The main outcome variable of this study was appropriate infant feeding practice. The primary independent variable was postpartum depression, and secondary independent variables were socio-demographic and maternal health care related characteristics such as place of delivery, mode of delivery, antenatal care utilization, and postnatal care.

## Data quality control

Data collectors and supervisors with prior experience were recruited and intensively trained for three days on the purpose of the study and how to collect data. The questionnaire was translated from English to Tigrigna (local language) and then translated back to English to

check its consistency. A pretest was done on 5% of mothers in the study area, which was not involved in the main study. During the pre-test, the questionnaire was assessed for its understandability, time needed to complete the interview, and for cultural acceptability in the area, and any necessary corrections were made.

## Data processing and analysis

Data were cleaned, coded, and entered into Epi-info version 7 and then transferred to SPSS version 23 for analysis. During analysis, a composite variable for appropriate infant feeding practice based on age was developed, which includes the sum of exclusive breastfeeding practices for infants aged 0 to 5 months and appropriate complementary feeding practices for infants aged 6 to 11 months. Descriptive analysis such as frequency tables, means, and standard deviations were used to describe the data. In bivariate analysis, crude odds ratio with a 95% confidence interval was calculated to measure the degree of association between independent variables and appropriate infant feeding practice. Variables with a p-value $\leq 0.25$ in bivariate analysis were included in the final model of multivariable logistic regression analysis. Finally, binary logistic regression was employed to assess the strength of the association between postpartum depression and appropriate infant feeding practice by controlling potential confounding variables. The significance level was declared at p-value $<0.05$. Hosmer and Lemeshow test was computed to check model fitness, which was not significant (p-value = 0.955).

## Operational definition

Appropriate infant feeding practice was defined as only exclusive breastfeeding for infants less than 6 months and the timely introduction of complementary feeding, minimum meal frequency, minimum dietary diversity, and minimum acceptable diet in infants aged 6–11 months.

Postpartum depression was defined as postpartum women with SRQ-20 score of six or above.

Exclusive breastfeeding was defined as infants who have received breastmilk only and no other liquids or solid foods, with the exception of vitamins or medicines in the first six months.

Appropriate complementary feeding was defined as infants 6–11 months of age who received a timely introduction of complementary feeding, minimum meal frequency, minimum dietary diversity, and minimum acceptable diet.

## Ethical consideration

Ethical approval was obtained from Addis Ababa University, health research and ethics committee (REC/0018/2019), and Mekele University granted a letter of permission to conduct the study. Informed verbal consent and assent were obtained from each sampled infant's mother. Before the beginning of data collection, adequate information was provided to study participants about the purpose of the study and the confidentiality of the information they provided during their participation, if they were willing. Finally, mothers were informed that they could refuse or withdraw from the study at any time.

## Results

### Socioeconomic-demographic characteristics

We retrieved complete data from a total of 513 mothers, yielding a response rate of 97.7%. The overall mean (SD) age of mothers was 31.1(4.8) years. One hundred forty-five (84.8%) of

mothers with PPD and 294 (85.9%) mothers without PPD were married at the time of the survey. Regarding mother's educational status, 38.5% and 46.2% of mothers with PPD and mothers without PPD had no formal education, respectively. The majority of the participants (58.1%) lived in the rural area. The overall median (IQR) monthly income was 1800 (1833) Ethiopian Birr. Seventy-six (44.4%) of households with postpartum depression and 47.4% of households without postpartum depression earned below the median level of income.

Of the total children enrolled in the study, 265 (51.7%) were males and 248 (48.3%) were females. The overall infant age mean (SD) was 5.61 (3.19) (13.9) months. The mean (SD) age of the immediate older child was 19.1(14.5) months for an exposed mother and 22.7(18) months for unexposed mothers. The proportion of infants with birth order of three or above was 12.7% (Table 1).

## Health care related characteristics

Almost all 160 (93.6%) of mothers with PPD and 322 (94.2%) of mothers without PPD had at least one antenatal care visit during their last pregnancy. More than half (61.4%) of mothers with PPD and 60.2% of mothers without PPD were multipara (2–4births). About 146 (85.4%) of depressed mothers and 295 (86.3%) of non-depressed mothers delivered their last child at a health facility. Seventy-three (42.7%) of mothers with PPD and 213 (62.3%) of mothers without PPD had received postnatal care after their last child delivery. The majority (95.7%) of the mothers answered that their current pregnancy status was planned (Table 2).

**Postpartum depression among study participants.** The overall mean (SD) score value (the number of yeses to the SRQ-20 questions) was 4.8(3.4), ranging between 0 and 17. The internal consistency of the SRQ-20 was assessed using the Cronbach's alpha, with a level of 0.703. The chi-square test revealed no significant difference between mothers with PPD and mothers without PPD in basic socio-demographic factors such as maternal education, maternal occupation, and marital status. The most commonly reported symptoms of postpartum depression in mothers were: "headache" (58.7%), followed by "easily tired" (50.9%), and "feeling tired all the time" (44.8%). Overall, 3.7% of them had suicidal ideation within the last 30 days prior to the survey.

## Magnitude of appropriate Infant feeding practice

The overall prevalence of appropriate infant feeding practice was 37.6% (95% CI: 33.5%-41.9%). The proportion of mothers without PPD who were practicing appropriate infant feeding practice was 42.7% (95% CI: 42.9%–53.2%), whereas only 27.5% (95% CI: 24.7%–32.5%) of mothers with PPD were practicing appropriate infant feeding.

**Magnitude of appropriate exclusive breastfeeding practice.** The proportion of appropriate exclusive breastfeeding was significantly higher among mothers without PPD, 59% (95% CI: 53.4%–64.6%) compared to mothers with PPD 46.2% (95% CI: 40.5%–51.9%) (p-value = 0.041).Receiving plain water was the main reason for inappropriate infant feeding practices below the age of 6 months, and it was statistically different between mothers with PPD and without PPD (p-value = 0.021) (S1 Fig).

**Magnitude of appropriate complementary feeding practice.** The magnitude of mothers without PPD who were practicing appropriate complementary feeding was 19.7% (95% CI: 13.5%–27.2%), whereas only 5.1% (95% CI: 1.4%–12.6%) of mothers with PPD were practicing appropriate complementary feeding. This difference was statistically significant (p-value = 0.03) (S2 Fig).

**Table 1. Socio-demographic characteristics of participants in eastern zone of Tigray, Ethiopia.** (N = 513).

| Characteristics | PPD | | Non PPD | | Total n (%) | | P -value |
|---|---|---|---|---|---|---|---|
| | Frequency | % | Frequency | % | Frequency | % | |
| Maternal age | | | | | | | 0.278 |
| < = 24 | 13 | 7.6 | 32 | 9.4 | 45 | 8.8 | |
| 25–29 | 53 | 31 | 87 | 25.4 | 140 | 27.3 | |
| 30–35 | 78 | 45.6 | 149 | 43.6 | 227 | 44.2 | |
| >35 | 27 | 15.8 | 74 | 21.6 | 101 | 19.7 | |
| Religion | | | | | | | .757 |
| Orthodox | 145 | 84.7 | 283 | 82.7 | 428 | 83.4 | |
| Muslim | 20 | 11.7 | 48 | 14 | 68 | 13.2 | |
| Others [a] | 6 | 3.5 | 11 | 3.3 | 17 | 3.3 | |
| Marital status | | | | | | | .898 |
| Married | 145 | 84.8 | 294 | 86 | 439 | 85.6 | |
| Single | 6 | 3.5 | 15 | 4.4 | 21 | 4.1 | |
| Divorced | 9 | 5.3 | 15 | 4.4 | 24 | 4.7 | |
| Widowed/separated | 11 | 6.4 | 18 | 5.2 | 29 | 5.6 | |
| Maternal educational status | | | | | | | .158 |
| No formal education | 66 | 38.5 | 158 | 46.2 | 224 | 43.7 | |
| Primary school | 59 | 34.5 | 84 | 24.6 | 143 | 27.9 | |
| Secondary and above | 46 | 26.9 | 100 | 29.2 | 146 | 28.4 | |
| Maternal Occupation | | | | | | | .055 |
| Housewife | 90 | 52.6 | 204 | 59.6 | 294 | 57.3 | |
| Farmer | 37 | 21.6 | 52 | 15.3 | 89 | 17.4 | |
| Merchant | 19 | 11.1 | 39 | 11.4 | 58 | 11.3 | |
| Gov't employee | 15 | 8.8 | 24 | 7 | 39 | 7.6 | |
| Others [b] | 10 | 5.9 | 23 | 6.7 | 33 | 6.4 | |
| Husband education | | | | | | | <0.001 |
| No formal education | 54 | 31.8 | 141 | 41.2 | 195 | 38 | |
| Primary school | 58 | 33.9 | 69 | 20.2 | 127 | 24.8 | |
| Secondary and above | 59 | 34.5 | 132 | 38.6 | 191 | 37.2 | |
| Place of residence | | | | | | | .129 |
| Urban | 79 | 46.2 | 136 | 39.8 | 215 | 41.9 | |
| Rural | 92 | 53.8 | 206 | 60.2 | 298 | 58.1 | |
| Monthly income | | | | | | | .083 |
| < = 999 ETB | 41 | 24 | 80 | 23.4 | 121 | 23.6 | |
| 1000–1999 ETB | 71 | 42.1 | 133 | 38.9 | 205 | 40 | |
| 2000–2999 ETB | 22 | 12.9 | 41 | 12 | 63 | 12.3 | |
| 3000–3999 ETB | 18 | 10.5 | 27 | 7.9 | 45 | 8.8 | |
| > = 4000 ETB | 18 | 10.5 | 61 | 17.8 | 79 | 15.3 | |
| Birth order | | | | | | | 0.925 |
| < = 3 | 149 | 87.1 | 299 | 87.4 | 448 | 87.3 | |
| >3 | 22 | 12.3 | 43 | 12.6 | 65 | 12.7 | |
| Infant age (months) | | | | | | | 0.377 |
| 0–5 | 93 | 54.4 | 200 | 58.5 | 293 | 57.1 | |
| 6–11 | 78 | 45.6 | 142 | 41.5 | 220 | 42.9 | |
| mean (SD) | 5.94 (3.01) | | 5.45 (3.26) | | 5.61 (3.19) | | |
| Paternal substance use | | | | | | | .001 |
| YES | 13 | 7.6 | 22 | 6.4 | 35 | 6.8 | |

(*Continued*)

**Table 1.** (Continued)

| Characteristics | PPD | | Non PPD | | Total n (%) | | P -value |
|---|---|---|---|---|---|---|---|
| | Frequency | % | Frequency | % | Frequency | % | |
| NO | 158 | 92.4 | 320 | 93.6 | 478 | 93.2 | |

[a] protestant, catholic; Gov't employee, Government employee

[b] student, unemployed; ETB, Ethiopian Birr, SD, standard deviation

**Factors associated with exclusive breastfeeding.** The result from the regression analysis showed that exclusive breastfeeding was more likely among mothers who had $\geq$4 ANC visits (AOR = 1.66; 95% CI: 1.03–2.69) than mothers who had ANC visits below four (Table 3).

**Factors associated with appropriate complementary feeding.** The result shows that the odds of appropriate complementary feeding among mothers without PPD were 3.27 times higher than those mothers with postpartum depression (AOR = 3.27; 95% CI: 1.2–8.9) (Table 4).

## Factors associated with appropriate infant feeding practices

In the bivariate analysis, postpartum depression was significantly associated with appropriate complementary feeding, exclusive breastfeeding, and appropriate infant feeding practice. The odds of appropriate infant feeding practice were 1.96 times higher among mothers without postpartum depression (COR = 1.96; 95% CI: 1.32–2.93). After adjusting for potential confounding factors, the odds of appropriate infant feeding practice were 2 times higher among mothers without postpartum depression compared to their counterparts (AOR = 2.03; 95% CI: 1.29–3.18).

Other variables such as monthly income and birth order were significantly associated with appropriate infant feeding practices. With regard to monthly income, mothers from

**Table 2. Maternal health care related characteristics in eastern zone of Tigray, Ethiopia.**

| Characteristics | PPD | | Non PPD | | Total N (%) | | p-value |
|---|---|---|---|---|---|---|---|
| | Frequency | % | Frequency | % | Frequency | % | |
| Place of delivery | | | | | | | .296 |
| Home | 25 | 14.6 | 47 | 13.7 | 72 | 14 | |
| Health facility | 146 | 85.4 | 295 | 86.3 | 441 | 86 | |
| ANC utilization | | | | | | | .250 |
| YES | 160 | 93.6 | 322 | 94.2 | 482 | 93.4 | |
| NO | 11 | 6.4 | 20 | 5.8 | 31 | 6.6 | |
| PNC utilization | | | | | | | <0.001 |
| YES | 73 | 42.7 | 213 | 62.3 | 286 | 55.7 | |
| NO | 98 | 57.3 | 129 | 37.7 | 227 | 44.3 | |
| Pregnancy status | | | | | | | <0.001 |
| Planned | 155 | 90.6 | 336 | 98.2 | 491 | 95.7 | |
| Unplanned | 16 | 9.4 | 6 | 1.8 | 22 | 4.3 | |
| Parity | | | | | | | .905 |
| Primipara | 69 | 39.7 | 105 | 60.3 | 174 | 34 | |
| Multipara | 118 | 37.9 | 193 | 62.1 | 311 | 60.2 | |
| Grand multipara | 6 | 21.4 | 22 | 78.6 | 28 | 5.8 | |

[*]Significant at P < 0.05

**Table 3. Factors associated with exclusive breastfeeding in eastern zone of Tigray, Ethiopia.**

| Characteristics | Exclusive breastfeeding | | COR (95% CI) | AOR (95% CI) |
|---|---|---|---|---|
| | Appropriate | Inappropriate | | |
| | N (%) | N (%) | | |
| **PNC utilization** | | | | |
| **No** | 73(32.2) | 154(67.8) | 1.00 | |
| **Yes** | 120(42) | 166(58) | 1.2(0.83–1.73) | 1.5(1.04–2.16) |
| **ANC visit** | | | | |
| <4 times | 84(31.2) | 185(68.8) | 1.00 | 1.00 |
| > = 4 times | 109(44.7) | 135(55.3) | 1.5(1.04–2.16) * | 1.66(1.03–2.69) * |

PNC postnatal care, ANC antenatal care

*Significant at P < 0.05

households that earn 1000–1999 ETB (AOR = 2.26; 95% CI: 1.02–5.01), 2000–2999 ETB (AOR = 1.96; 95% CI: 1.21–4.73) and 3000–3999 ETB (AOR = 5.31; 95% CI: 1.97–13.4) had higher odds of appropriate infant feeding practice. The odds of appropriate infant feeding practices among infants with a birth order above three was 58% (AOR = 0.42; 95% CI: 0.26–0.97) less than those infants with a birth order of three and below (Table 5).

## Discussion

Safe and adequate infant feeding practices are critical for reducing malnutrition in developing countries like Ethiopia. As a result, WHO recommended basic infant feeding practices. This study assessed the impact of postpartum depression on infant feeding practices in the eastern zone of Tigray. Postpartum depression, households with higher monthly income, and birth order of infants above three were independent predictors of appropriate infant feeding practice.

In our study, the overall prevalence of infant feeding practice was 37.6%, which was much higher than the national mini-health surveys reported in 2019 (9.76%) [21]. Likewise, this finding is higher than the previous studies conducted in different parts of Ethiopia: Bahir Dar city (7%), Arsi Negele Woreda (9.5%), Horro district in western Ethiopia (9.91%), and Shashemene Woreda (32.1%), respectively [22–25]. The possible explanation for the difference could be due to variations in maternal sociodemographic characteristics, cross-cultural differences in infant feeding practice, and access to maternal health care services. Another possible explanation could be due to the time gap between studies. However, the finding was lower when compared to studies conducted in North Showa (65.8%) [26], and Kalu district, Northeast Ethiopia (57.7%) [27]. This might be due to the high illiteracy rate compared to other study sites.

**Table 4. Factors associated with appropriate complementary feeding in eastern zone of Tigray, Ethiopia.**

| Characteristics | Complementary feeding practice | | COR (95% CI) | AOR (95% CI) |
|---|---|---|---|---|
| | Appropriate | Inappropriate | | |
| | N (%) | N (%) | | |
| **Depression status** | | | | |
| **Postpartum depressed** | 47(27.5) | 124(72.5) | 1.00 | 1.00 |
| **Non postpartum depressed** | 146(42.7) | 196(57.3) | 2.73(1.03–7.24) * | 3.27 (1.2–8.9) * |

*Significant at P < 0.05

**Table 5. Independent factors associated with appropriate infant feeding practice in eastern zone of Tigray, Ethiopia.**

| Characteristics | Infant feeding practice | | COR (95% CI) | AOR (95% CI) |
|---|---|---|---|---|
| | Appropriate | Inappropriate | | |
| | N (%) | N (%) | | |
| Depression status | | | | |
| Postpartum depressed | 47(27.5) | 124(72.5) | 1.00 | 1.00 |
| Non postpartum depressed | 146(42.7) | 196(57.3) | 1.96(1.32–2.93) | 2.03 (1.29–3.18) * |
| Marital status | | | | |
| Married | 171 (39) | 268 (61) | 1.00 | 1.00 |
| Divorced/separated | 14 (34.1) | 27(65.9) | 0.81(0.41–1.59) | 1.78(0.62–5.16) |
| Others [a] | 8 (24.2) | 25 (75.8) | 0.50 (0.22–1.14) | 2.17(0.93–5.08) |
| Maternal occupation | | | | |
| Housewife | 144(38.8) | 180 (61.2) | 1.00 | 1.00 |
| Farmer | 24(29.3) | 58 (70.7) | 0.65(0.38–1.11) | 0.74(0.42–1.29) |
| Merchant | 27 (46.6) | 31 (53.4) | 1.38(0.78–2.42) | 1.37(0.76–2.48) |
| Government employee | 11 (29.7) | 26 (70.3) | 0.67(0.32–1.40) | 0.77(0.35–1.68) |
| Others [b] | 17 (40.5) | 25(59.5) | 1.07 (0.56–2.08) | 1.24(0.62–2.47) |
| Monthly income | | | | |
| < = 999 ETB | 41(24.0) | 80(23.4) | 1.00 | 1.00 |
| 1000–1999 ETB | 71(42.1) | 133(38.9) | 2.12(0.97–4.64) | 2.26(1.01–5.01) * |
| 2000–2999 ETB | 22(12.9) | 41(12.0) | 1.90(1.01–4.48) * | 1.96(1.21–4.73) * |
| 3000–3999 ETB | 18(10.5) | 27(7.9) | 4.57(1.79–11.6) * | 5.13(1.97–13.4) * |
| > = 4000 ETB | 18(10.5) | 61(17.8) | 2.21(0.94–5.22) | 2.22(0.93–5.33) |
| **Infant age** (months) | | | | |
| **0–5** | 161(54.9) | 132(45.1) | 1.00 | 1.00 |
| **6–11** | 32(14.5) | 188(85.5) | 0.14(0.09–0.22) | 0.13(0.08–0.2) |
| ANC utilization | | | | |
| No | 14(43.8) | 18(56.3) | 1.00 | 1.00 |
| Yes | 179(37.2) | 302(62.8) | 0.94(0.49–1.80) | 0.68(0.34–1.38) |
| Birth order | | | | |
| < = 3 | 177(39.5) | 271 (60.5) | 1.00 | 1.00 |
| >3 | 16 (24.6) | 49 (75.4) | 0.50 (0.28–0.91) | 0.42(0.26–0.97) * |
| Parity | | | | |
| Primipara | 69(39.7) | 105(60.3) | 1.00 | 1.00 |
| Multipara | 118(37.9) | 193(62.1) | 0.93(0.64–1.36) | 1.05(0.70–1.60) |
| Grand multipara | 6(21.4) | 22(78.6) | 0.42(0.16–1.08) | 0.73 (0.20–2.63) |
| Postnatal care | | | | |
| No | 73(32.2) | 154(67.8) | 1.00 | 1.00 |
| Yes | 120(42) | 166(58) | 1.52(1.06–2.20) * | 1.02(0.57–1.76) |
| ANC visit | | | | |
| <4 times | 84(31.2) | 185(68.8) | 1.00 | 1.00 |
| > = 4 times | 109(44.7) | 135(55.3) | 1.78 (1.24–2.55) * | 0.83 (0.44–1.57) |

COR Crude odds ratio, AOR Adjusted odds ratio, CI Confidence interval

[a] protestant, catholic,

[b] student, unemployed

*Significant at P < 0.05

Mothers with little or no education can easily be influenced by others such as peers, parents, and untrained birth attendants to practice inappropriate infant feeding.

This study demonstrated that postpartum depression is significantly associated with infant feeding practice. Similarly, a qualitative systematic review conducted in developed countries shows that women with early postpartum depression had a negative influence on infant feeding outcomes [12], and a study report from low and middle-income countries shows that depressive symptoms have been associated with a short duration of breastfeeding and mothers with depressive symptoms in the first 4–6 weeks postpartum were more likely to stop breastfeeding earlier than non-depressed mothers [28]. The possible justification could be that postpartum depression may negatively interfere with maternal self-esteem and self-confidence. Moreover, postpartum depression impairs the mother's interaction with her infant, such as feeding, touch, and sensitivity, which increases breastfeeding difficulties and reduces appropriate infant feeding.

This study was consistent with previous results conducted in Canada that showed mothers with postpartum depression at 3 months were associated with an 11% reduction in the odds of exclusive breastfeeding at 6 months [11]. In Brazil, a higher risk of early interruption of exclusive breastfeeding among mothers with postpartum depressive symptoms was reported [29], and in a subsequent study, women with postpartum depression were more likely to practice nonexclusive breastfeeding [30]. These findings clearly demonstrate that mothers with postpartum depression are less confident in their ability to breastfeed. This implies the need for early detection and effective treatment of postpartum depression in primary health care to reduce the potentially adverse effects on the mother and child. However, an inverse relationship was found in a report from northern Ghana. According to this report, there is no significant association between maternal depression and complementary feeding indicators [31]. The possible reasons could be due to using the locally invalidated tool for assessing maternal postpartum depression and the small sample size, which reduces the power of the study and makes it difficult to find the association between postpartum depression and complementary feeding practice.

The current study determined that monthly income was significantly associated with appropriate infant feeding practices. Mothers from households with a higher monthly income had higher odds of practicing appropriate infant feeding than mothers with lower monthly income. Similar to a finding reported in another study [32], mothers with higher monthly incomes are more likely to practice infant feeding appropriately. This may be due to the fact that mothers with high incomes tend to spend more time feeding their infants appropriately.

This study also revealed that birth order was found to be a significant predictor of appropriate infant feeding practice, which is congruent with the findings from Damot [33]. The possible explanation is that mothers with higher birth order infants bear a greater burden in caring for their children, making the mother unable to devote more time to proper infant feeding. Another possible justification could be due to food insecurity in a large family.

## Limitation of the study

The limitation of this study is the possible inability to assess a causality/temporal relationship between postpartum depression and feeding practices for children younger than one year. However, the knowledge acquired from this research enabled us to describe the baseline information for future comparisons and/or discussions. Besides, some variables were assessed by the mother's self-report, which mainly depends on recall; this may lead to recall bias.

## Conclusion

The overall prevalence of appropriate infant feeding practice was low in the study area. This study shows postpartum depression as an important predictor of appropriate infant feeding

practices. Infants with a birth order of three or above and mothers with higher household monthly income were identified as significant predictors of appropriate infant feeding practice. Therefore, early recognition, screening, and raising awareness of postpartum depression symptoms should be considered to promote appropriate infant feeding practices. Moreover, policymakers need to strengthen the provisions for nutrition education while providing economic support to households with low income, health education for multiparous women, and incorporating maternal mental health with routine maternal health care services.

## Supporting information

**S1 Fig. The proportion of exclusive breastfeeding among exposed and unexposed mothers, Ethiopia.**
(TIF)

**S2 Fig. The proportion of complementary feeding among exposed and unexposed mothers in Ethiopia.**
(TIF)

**S1 File.**
(SAV)

**S1 Questionnaire.**
(DOCX)

## Acknowledgments

We would like to appreciate Mekele University for their technical support and cooperation in writing support letters to the concerned bodies and/or facilitating data collection. We are grateful to Kilite Awlaelo HDSS coordinating office for providing us with the required data. We also thank the data collectors and supervisors for their hard work. Finally, our special thanks go to the study participants; without their participation, this paper would not be possible.

## Author Contributions

**Conceptualization:** Angesom Weldu, Mengistu Yilma.

**Data curation:** Angesom Weldu, Ayele Belachew, Mengistu Yilma.

**Formal analysis:** Angesom Weldu, Ayele Belachew, Mengistu Yilma.

**Investigation:** Angesom Weldu.

**Methodology:** Angesom Weldu, Ayele Belachew, Mengistu Yilma.

**Software:** Angesom Weldu, Ayele Belachew, Mengistu Yilma.

**Supervision:** Ayele Belachew, Mengistu Yilma.

**Validation:** Angesom Weldu, Ayele Belachew, Mengistu Yilma.

**Visualization:** Angesom Weldu, Ayele Belachew, Mengistu Yilma.

**Writing – original draft:** Angesom Weldu.

**Writing – review & editing:** Angesom Weldu, Ayele Belachew.

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
