## [Decision Letter · Decision Letter 0]

18 Mar 2021

PONE-D-21-02467

THE RELATIONSHIP BETWEEN POSTPARTUM DEPRESSION AND APPROPRIATE INFANT FEEDING PRACTICE IN EASTERN ZONE OF TIGRAY, ETHIOPIA: COMPARATIVE CROSS-SECTIONAL STUDY

PLOS ONE

Dear Dr. Weldu,

Thank you for submitting your manuscript to PLOS ONE. After careful consideration, we feel that it has merit but does not fully meet PLOS ONE’s publication criteria as it currently stands. Therefore, we invite you to submit a revised version of the manuscript that addresses the points raised during the review process.

We look forward to receiving your revised manuscript.

Kind regards,

Claudia Marotta

Academic Editor

PLOS ONE

Additional Editor Comments:

dear Authors follow reviewer suggestions to improve your paper.

Journal Requirements:

2. Please include your tables as part of your main manuscript and remove the individual files. Please note that supplementary tables should be uploaded as separate "supporting information" files.

3. Please provide additional details regarding participant consent. In the ethics statement in the Methods and online submission information, please ensure that you have specified whether consent was written or verbal/oral. If consent was verbal/oral, please specify: 1) whether the ethics committee approved the verbal/oral consent procedure, 2) why written consent could not be obtained, and 3) how verbal/oral consent was recorded. If your study included minors, please state whether you obtained consent from parents or guardians in these cases. If the need for consent was waived by the ethics committee, please include this information.

"The funders had no role in study design, data collection and analysis, decision to publish, or preparation of the manuscript"

7. Please ensure that you refer to Figure 1 in your text as, if accepted, production will need this reference to link the reader to the figure.

8. We note you have included a table to which you do not refer in the text of your manuscript. Please ensure that you refer to Tables 1-3 in your text; if accepted, production will need this reference to link the reader to the Tables.

Reviewers' comments:

Reviewer's Responses to Questions

**Comments to the Author**

1. Is the manuscript technically sound, and do the data support the conclusions?

Reviewer #1: Yes

Reviewer #2: Yes

2. Has the statistical analysis been performed appropriately and rigorously? 

Reviewer #1: Yes

Reviewer #2: No

3. Have the authors made all data underlying the findings in their manuscript fully available?

Reviewer #1: Yes

Reviewer #2: Yes

4. Is the manuscript presented in an intelligible fashion and written in standard English?

Reviewer #1: No

Reviewer #2: Yes

5. Review Comments to the Author

Reviewer #1: Thank you for addressing an important topic. It is great to highlight the impact of PPD on infant feeding practices and such articles should also highlight the need to reduce the stigma associated with mental health. It would be good to add a few sentences on some background on the extent of Postpartum depression globally and the extent of underreporting. In addition, would be good to highlight the impact on women themselves.

In the abstract, within the conclusion the authors write -"this study showed low prevalence and postpartum depression"- it is not clear what has low prevalence. Please state where PPD is linked with poor feeding practices. What about other children in the family- do they have a similar fate?

Please break your sentences and add the appropriate source according for the sentence. In lines 56-61, it is unclear which source is relevant for what statistics you are quoting.

The description of the mothers in lines 195-205 could focus on bringing out the differences between the mothers who had PPD and those who did not. The description is hard to go through. Maybe better to summarize the similarities and focus more on the differences.

Is there a linkage between any history of depression in previous pregnancies and PPD in future? Is that important and was that asked?

Check line 211. Incomplete sentence?

The discussion on birth order is not clearly stated. Please revise lines 287-295 to more clearly articulate your findings first.

The authors can strengthen the policy and program recommendations for PPD and its impact. Early recognition, raising awareness of symptoms associated with PD, eliminating stigma, involving the extended household and community to support women going through PPD so infant feeding practices are not impacted are some considerations.

A strong edit of the entire article is needed.

Reviewer #2: I found this article very interesting, but some changes should be made. There are some typo errors, specially missing spaces in text. I find the terms "exposed and unexposed" confusing and would advise to change them.

Abstract:

-rewrite the first sentence of the conclusion for clarity "this study showed low prevalence and postpartum depression as an important contributor to the appropriate infant feeding practice".

Introduction:

- line 76: there is a typo, with a point before the word "studies" that should be corrected: "Ethiopia [9,10] .Studies"

Statistics:

-Control variables must be specified

Results:

-line 189 typo error, capital letter that should not be there in "retrieved Complete data".

- the sociodemographic variables results should be compared between groups through statistic tests (chi squared, T test...) to assess if statistically significant differences arise. This information should be included in the final tables, and it should be decided which data to include in the text and which in the table, so the text would be more easily read.

- it would be very important to describe, if possible, why the feeding was not appropriate, e.g. use of other foods before 6 months, interruption of breastfeeding, lack of variety in the diet, etc. This could help to have a better understanding of the reasons why an appropriate feeding practice do not happen (and what measures could be implemented to improve this practice).

Discussion:

-line 255: clarify this sentence

- line 269: rewrite this sentence

- line 292: correct typo ".this"

- have the authors considered that the most frequent depressive symptoms described (headache, being tired) could be related to the difficulties inherently tied to breastfeeding? Could it be therefore a confusion between the consequences of breastfeeding and depression?

- again, to know exactly what was the reason for inappropriate feeding of the infants would be of the utmost importance to provide an interpretation of the difficulties mothers and particularly the ones suffering from PPD face.

Tables: all abbreviations used in the table should be clarified above.

6. PLOS authors have the option to publish the peer review history of their article (what does this mean?). If published, this will include your full peer review and any attached files.

Reviewer #1: No

Reviewer #2: **Yes: **Monica Sanchez-Autet

---

## [Author Response · Author response to Decision Letter 0]

9 May 2021

Dear Editors,

We greatly appreciate the time and effort put forth by reviewers and editors to improve our paper, ‘‘The relationship between postpartum depression and appropriate infant feeding practice in Eastern zone of Tigray, Ethiopia: A comparative cross-sectional study” which we have addressed below. If any responses are unclear or you wish additional changes, please let us know.

Editor comments 

Comment 1: Please ensure that your manuscript meets PLOS ONE's style requirements, including those for file naming.

Author Response: The manuscript has been revised intensively in line with the journal’s requirements. Therefore, the title has changed to sentence case (Page 1) and asterixis symbol used in corresponding author (line 15 of page 1) and added a city after the institution in the affiliation part (line 10 of page 1).

Comment 2: Please include your tables as part of your main manuscript and remove the individual files. Please note that supplementary tables should be uploaded as separate "supporting information" files.

Author Response: We have also modified the figure cite right next to the paragraph represented and removed from main manuscript (line 253-254 of page 15) and uploaded as separate file.

Comment 3. Please provide additional details regarding participant consent. In the ethics statement in the Methods and online submission information, please ensure that you have specified whether consent was written or verbal/oral. If consent was verbal/oral, please specify: 

1) whether the ethics committee approved the verbal/oral consent procedure,

 2) why written consent could not be obtained, and

 3) how verbal/oral consent was recorded. If your study included minors, please state whether you obtained consent from parents or guardians in these cases. If the need for consent was waived by the ethics committee, please include this information

 Author Response: We have agreed and incorporated additional details about participants consent. 

 1) Informed verbal consent was obtained after approval by the research ethical committee number (REC/0018/2019).

 2) Since national and regional statistics show the nation’s (Ethiopia) rural and semi urban areas which account the large proportion of geographical area of the country have high rate of illiteracy that makes difficult to obtain written consent from participants. In addition, our study neither involve with sensitive issue nor uses any procedure to adversely affect participants welfare.

 3) After reading the whole information about the purpose, risk, beneficence and non-maleficence, participants were asked whether they are willing to participate in the study with a statement 

“Do you give your permission for me to interview you?”

 A) Yes, I give you my permission 

 B) No, I am not willing to participate 

If he/she agrees to participate we tick on the box. 

informed verbal consent was recorded However, we believe that the minors (infants in our case) are not directly involved in the study only the mothers provide the information by remembering the past feeding practice. 

Comment 4: Thank you for stating the following financial disclosure: "The funders had no role in study design, data collection and analysis, decision to publish, or preparation of the manuscript" At this time, please address the following queries:

A) Please clarify the sources of funding (financial or material support) for your study. List the grants or organizations that supported your study, including funding received from your institution.

Author response: The funding source of this was Addis Ababa University. Amount received 25,000 Ethiopian Birr.

B) State what role the funders took in the study. If the funders had no role in your study, please state: “The funders had no role in study design, data collection and analysis, decision to publish, or preparation of the manuscript.”

Author response: “The funders had no role in study design, data collection and analysis, decision to publish, and preparation of the manuscript.”

C) If any authors received a salary from any of your funders, please state which authors and which funders. Please include your amended statements within your cover letter; we will change the online submission form on your behalf.

Comment 5. We note that you have stated that you will provide repository information for your data at acceptance. Should your manuscript be accepted for publication, we will hold it until you provide the relevant accession numbers or DOIs necessary to access your data. If you wish to make changes to your Data Availability statement, please describe these changes in your cover letter and we will update your Data Availability statement to reflect the information you provide.

Author response: Thank you for the suggestion. I will make the necessary data available as per your request.

Comment 6. PLOS requires an ORCID iD for the corresponding author in Editorial Manager on papers submitted after December 6th, 2016. Please ensure that you have an ORCID iD and that it is validated in Editorial Manager.

Author response: I do have ORCID iD and I already validated in PLOS editorial manager. 

Comment 7: Please ensure that you refer to Figure 1 in your text as, if accepted, production will need this reference to link the reader to the figure.

Author response: I have corrected the figure 1 cite to Fig 1 and put it in separate file (line 253 of page 15) 

Comment 8. We note you have included a table to which you do not refer in the text of your manuscript. Please ensure that you refer to Tables 1-3 in your text; if accepted, production will need this reference to link the reader to the Tables

Author response: Thank you for the comment. we have corrected it accordingly. 

Reviewers' comments:

Reviewer's Responses to Questions

Comments to the Author

1. Is the manuscript technically sound, and do the data support the conclusions?

Reviewer #1: Yes

Reviewer #2: Yes

2. Has the statistical analysis been performed appropriately and rigorously?

Reviewer #1: Yes

Reviewer #2: No

3. Have the authors made all data underlying the findings in their manuscript fully available?

Reviewer #1: Yes

Reviewer #2: Yes

4. Is the manuscript presented in an intelligible fashion and written in standard English?

Reviewer #1: No

Reviewer #2: Yes

5. Review Comments to the Author

Reviewers' comments:

Reviewer #1: Thank you for addressing an important topic. It is great to highlight the impact of PPD on infant feeding practices and such articles should also highlight the need to reduce the stigma associated with mental health. It would be good to add a few sentences on some background on the extent of Postpartum depression globally and the extent of underreporting. In addition, would be good to highlight the impact on women themselves.

Author response: we greatly appreciate for the compliment and constructive comments. We have included global burden of Postpartum depression and the effect of Postpartum depression on mothers and infants on background.

In the abstract, within the conclusion the authors write -"this study showed low prevalence and postpartum depression"- it is not clear what has low prevalence. Please state where PPD is linked with poor feeding practices. What about other children in the family- do they have a similar fate?

Author response: thank you for your view, it has been revised and stated in conclusion part as “This study showed low prevalence of appropriate infant feeding practice”. PPD is statistically linked with reduction of appropriate feeding practice.

Yes, we believe children in the same family will have similar fate because the unrecognized and untreated PPD results in reduced functioning of mother emotionally and physically which in turn have a negative influence of mother-infant interaction. Moreover, the persistent effect of untreated PPD will account for inability to take care of children in the family. 

Please break your sentences and add the appropriate source according for the sentence. In lines 56-61, it is unclear which source is relevant for what statistics you are quoting.

Author response: Thank you for the view, we break the sentences and cited appropriately it is revised as follows “…….young child feeding practice is attributed to 45% of all child deaths in which two thirds of those deaths occurred in the first year of life [2]”

The description of the mothers in lines 195-205 could focus on bringing out the differences between the mothers who had PPD and those who did not. The description is hard to go through. Maybe better to summarize the similarities and focus more on the differences.

Author response: Thank you for your suggestion, we have tried to focus on variables which shows significant difference using Chi square and factors that are believed to provide image about study participants to the reader were summarized. 

Is there a linkage between any history of depression in previous pregnancies and PPD in future? 

Author response: Yes, there is an association between any history of depression in previous pregnancies and PPD but sometimes can occur independently. 

Is that important and was that asked?

Author response: Thank you for your great question, Not asked because most literatures has been suggested that many of the postpartum mood disorders may actually begin during pregnancy or that antepartum depression increases the risk of developing PPD. It’s inevitable for a woman with antenatal depression/anxiety to have PPD and vice versa, mothers with PPD had experience antenatal depression/ anxiety. The other issue for not asking is we made decision not to introduce bias (recall bias while asking previous history of depression) 

Check line 211. Incomplete sentence?

Author response: revised.

The discussion on birth order is not clearly stated. Please revise lines 287-295 to more clearly articulate your findings first.

The authors can strengthen the policy and program recommendations for PPD and its impact. Early recognition, raising awareness of symptoms associated with PD, eliminating stigma, involving the extended household and community to support women going through PPD so infant feeding practices are not impacted are some considerations.

A strong edit of the entire article is needed.

Reviewer #2: I found this article very interesting, but some changes should be made. There are some typo errors, specially missing spaces in text. I find the terms "exposed and unexposed" confusing and would advise to change them.

Author response: we have revised it intensively and correct all suggested typo error in the manuscript and changed exposed into mothers with PPD and unexposed changed mothers without PPD throughout the paper.

Abstract:

-rewrite the first sentence of the conclusion for clarity "this study showed low prevalence and postpartum depression as an important contributor to the appropriate infant feeding practice".

Author response: Thank you for the view, we have revised and stated in conclusion as follows “……This study showed low prevalence of appropriate infant feeding practice…” 

Introduction:

- line 76: there is a typo, with a point before the word "studies" that should be corrected: "Ethiopia [9,10] .Studies"

Author response: revised and corrected.

Statistics:

-Control variables must be specified

Author response: The control variable are mothers without PPD.

Results:

-line 189 typo error, capital letter that should not be there in "retrieved Complete data".

Author response: revised and corrected.

- the sociodemographic variables results should be compared between groups through statistic tests (chi squared, T test...) to assess if statistically significant differences arise. This information should be included in the final tables, and it should be decided which data to include in the text and which in the table, so the text would be more easily read.

Author response: Revised and corrected

- it would be very important to describe, if possible, why the feeding was not appropriate, e.g. use of other foods before 6 months, interruption of breastfeeding, lack of variety in the diet, etc. This could help to have a better understanding of the reasons why an appropriate feeding practice do not happen (and what measures could be implemented to improve this practice).

Author response: Thank you for your suggestion. The appropriateness of feeding practice depends on age category of infant, mothers/caregiver with PPD give plain water frequently to infants less than 6 months of age and it was statistically significant(P-value-0.021) this was the main reason for inappropriateness of feeding practice among < 6 month of age, and the second important reason was receiving other liquids such as soup and broth before the age of six month. For infants 6-11 months of age, the main reason for inappropriate feeding was difficulty to receive minimum acceptable diet which was statistically significant (P-value = 0.04) different between mothers with PPD and mothers without PPD. 

Discussion:

-line 255: clarify this sentence

Author response: Sentence clarified 

- line 269: rewrite this sentence

Author response: Corrected 

- line 292: correct typo ".this"

Author response: Corrected 

- have the authors considered that the most frequent depressive symptoms described (headache, being tired) could be related to the difficulties inherently tied to breastfeeding? Could it be therefore a confusion between the consequences of breastfeeding and depression?

Author response: We have tried to put this temporality issue as a limitation of the study and must be cautiously interpreted as it would not enough to determine the causal association.in a specific way, whether PPD causes feeding difficulties or having feeding difficulty causes the mother to have depressive symptoms is mere possible to attain with such studies. Hence, we recommend further studies to be well designed prospective study to see the association between PPD and feeding difficulties or vice versa. 

- again, to know exactly what was the reason for inappropriate feeding of the infants would be of the utmost importance to provide an interpretation of the difficulties mothers and particularly the ones suffering from PPD face.

Author response: Thank you for your great view, mothers/caregiver with PPD give plain water frequently to infants less than 6 months of age and it was statistically significant (P-value-0.021) this was the main reason for inappropriateness of feeding practice among < 6 month of age, and the second important reason was receiving other liquids such as soup and broth before the age of six month. For infants 6-11 months of age, the main reason for inappropriate feeding was difficulty to receive minimum acceptable diet which was statistically significant (P-value = 0.04) different between mothers with PPD and mothers without PPD.

Tables: all abbreviations used in the table should be clarified above.

 Author response: revised and corrected. we have word abbreviated as Gov’t in to government in the table

---

## [Decision Letter · Decision Letter 1]

19 Jan 2022

PONE-D-21-02467R1THE RELATIONSHIP BETWEEN POSTPARTUM DEPRESSION AND APPROPRIATE INFANT FEEDING PRACTICE IN EASTERN ZONE OF TIGRAY, ETHIOPIA: A COMPARATIVE CROSS-SECTIONAL STUDYPLOS ONE

Dear Dr. Weldu,

Thank you for submitting your manuscript to PLOS ONE. After careful consideration, we feel that it has merit but does not fully meet PLOS ONE’s publication criteria as it currently stands. Therefore, we invite you to submit a revised version of the manuscript that addresses the points raised during the review process. Please carefully review abstract and write up of conclusions so ensure clarity and alignment with results from study. 

We look forward to receiving your revised manuscript.

Kind regards,

Melissa F. Young, Ph.D.

Academic Editor

PLOS ONE

Journal Requirements:

Reviewers' comments:

Reviewer's Responses to Questions

**Comments to the Author**

1. If the authors have adequately addressed your comments raised in a previous round of review and you feel that this manuscript is now acceptable for publication, you may indicate that here to bypass the “Comments to the Author” section, enter your conflict of interest statement in the “Confidential to Editor” section, and submit your "Accept" recommendation.

Reviewer #1: (No Response)

Reviewer #2: All comments have been addressed

2. Is the manuscript technically sound, and do the data support the conclusions?

Reviewer #1: Yes

Reviewer #2: Yes

3. Has the statistical analysis been performed appropriately and rigorously? 

Reviewer #1: Yes

Reviewer #2: Yes

4. Have the authors made all data underlying the findings in their manuscript fully available?

Reviewer #1: Yes

Reviewer #2: Yes

5. Is the manuscript presented in an intelligible fashion and written in standard English?

Reviewer #1: Yes

Reviewer #2: Yes

6. Review Comments to the Author

Reviewer #1: Thank you for addressing the reviewer feedback. Please revise the language in your conclusion from the abstract. There seems to be some confusion

This study showed low prevalence of appropriate infant feeding practice and significantly higher proportion of mother without postpartum depression practice appropriate infant feeding than their counterpart.

please add some language at the end about the low (overall) prevalence of appropriate infant feeding practices. While it is significantly lower in mothers with PPD, 42% is also very low in mothers without PPD. Please include that in your policy recommendations

Reviewer #2: I consider the revision adequate and the results interesting to be published. The subject is simple but relevant to mother and child health, therefore my advice would be to publish it.

7. PLOS authors have the option to publish the peer review history of their article (what does this mean?). If published, this will include your full peer review and any attached files.

Reviewer #1: No

Reviewer #2: No

---

## [Author Response · Author response to Decision Letter 1]

17 Feb 2022

response to Editor’s comment

Comment 1: Please carefully review abstract and write up of conclusions so ensure clarity and alignment with results from study.

Author response: Thank you for your interesting insight and suggestion. We have thoroughly reviewed our results, discussion and conclusion and the following statements were corrected. 

On conclusion part of abstract, we rephrased the statement with more sensible and aligned sentences with the write up. Now the statements read as follows “The overall prevalence of appropriate infant feeding practice in the study area was low. The prevalence of appropriate infant feeding practice was significantly different between mothers without postpartum depression and mothers with postpartum depression”.

journal requirement responses

Author response: Thank you for your constructive comments and suggestions. We took some time and checked the reference list and made some adjustments to make sure the bibliography is full and current to meet the journal requirements. None of the papers cited in the manuscript has retracted but only references number 8 has erratum and cited as per Plose One referencing style for original article with erratum.

response Reviewer 1 comments:

Author response: Thank you for your interesting insight, we have carefully revised and correct the statement as follows. This statement was rephrased “The overall prevalence of appropriate infant feeding practice in the study area was low. The prevalence of appropriate infant feeding practice was significantly different between mother without postpartum depression and mother with postpartum depression”. 

 instead of 

This study showed low prevalence of appropriate infant feeding practice and significantly higher proportion of mother without postpartum depression practice appropriate infant feeding than their counterpart.

Author response: We have included the word “overall” the overall prevalence of appropriate infant feeding practices was low.

Authors response: Thanks for your deep insight and we have made some recommendation for strengthening nutritional education to all mothers as appropriate infant feeding is still low in women without PPD. We added this recommendation to policy makers and it was very helpful. “Therefore, the modified statement reads as “strengthening the provisions of nutritional education, providing economic support to mothers with low income, health educating for multiparous and integrating maternal mental health with maternal health care services”. in addition, we recommend the policy makers to consider women with low income and birth order above three as it mentioned in the above phrase.

---

## [Editor Report · Decision Letter 2]

5 Apr 2022

PONE-D-21-02467R2THE RELATIONSHIP BETWEEN POSTPARTUM DEPRESSION AND APPROPRIATE INFANT FEEDING PRACTICE IN EASTERN ZONE OF TIGRAY, ETHIOPIA: A COMPARATIVE CROSS-SECTIONAL STUDYPLOS ONE

Dear Dr. Weldu,

Thank you for submitting your manuscript to PLOS ONE. After careful consideration, we feel that it has merit but does not fully meet PLOS ONE’s publication criteria as it currently stands. Therefore, we invite you to submit a revised version of the manuscript that addresses the points raised during the review process. Few minor corrections required. In abstract, please clearly define appropriate infant feeding practices in methods.  Include results for income in abstract results.

We look forward to receiving your revised manuscript.

Kind regards,

Melissa F. Young, Ph.D.

Academic Editor

PLOS ONE
---

## [Author Response · Author response to Decision Letter 2]

10 Apr 2022

editors comment: Few minor corrections required. In abstract, please clearly define appropriate infant feeding practices in methods. Include results for income in abstract results.

Author response: Thank you for your insightful comments. 

In the methods part of the abstract, we have changed infant feeding practice into appropriate infant feeding practice and restated as “Appropriate infant feeding practice was assessed using structured questionnaire obtained from Monitoring and Evaluating Breastfeeding Practices toolkit. 

In the result section of the abstract, we have added household monthly income as follows “Households with monthly income 1000-1999 ETB (AOR= 2.26; 95% CI: 1.01-5.08), 2000-2999 ETB (AOR= 1.96; 95% CI: 1.21-4.73) and 3000-3999 ETB (AOR= 5.13; 95% CI: 1.97-13.4) were more likely to practice appropriate infant feeding”.

---

## [Decision Letter · Decision Letter 3]

10 Aug 2022

PONE-D-21-02467R3THE RELATIONSHIP BETWEEN POSTPARTUM DEPRESSION AND APPROPRIATE INFANT FEEDING PRACTICE IN EASTERN ZONE OF TIGRAY, ETHIOPIA: A COMPARATIVE CROSS-SECTIONAL STUDYPLOS ONE

Dear Dr. Weldu,

Thank you for submitting your manuscript to PLOS ONE. After careful consideration, we feel that it has merit but does not fully meet PLOS ONE’s publication criteria as it currently stands. Therefore, we invite you to submit a revised version of the manuscript that addresses the points raised during the review process.

We look forward to receiving your revised manuscript.

Kind regards,

Md. Abdur Rafi

Academic Editor

PLOS ONE

Journal Requirements:

Please adjust the reviewers' comments.

Reviewers' comments:

Reviewer's Responses to Questions

**Comments to the Author**

1. If the authors have adequately addressed your comments raised in a previous round of review and you feel that this manuscript is now acceptable for publication, you may indicate that here to bypass the “Comments to the Author” section, enter your conflict of interest statement in the “Confidential to Editor” section, and submit your "Accept" recommendation.

Reviewer #1: All comments have been addressed

Reviewer #3: (No Response)

2. Is the manuscript technically sound, and do the data support the conclusions?

Reviewer #1: (No Response)

Reviewer #3: Yes

3. Has the statistical analysis been performed appropriately and rigorously? 

Reviewer #1: (No Response)

Reviewer #3: I Don't Know

4. Have the authors made all data underlying the findings in their manuscript fully available?

Reviewer #1: (No Response)

Reviewer #3: Yes

5. Is the manuscript presented in an intelligible fashion and written in standard English?

Reviewer #1: (No Response)

Reviewer #3: No

6. Review Comments to the Author

Reviewer #1: (No Response)

Reviewer #3: (No Response)

7. PLOS authors have the option to publish the peer review history of their article (what does this mean?). If published, this will include your full peer review and any attached files.

Reviewer #1: No

Reviewer #3: No

---

## [Author Response · Author response to Decision Letter 3]

24 Sep 2022

Dear Editors,

We greatly appreciate the time and effort put forth by reviewers and editors to improve our paper, ‘‘The relationship between postpartum depression and appropriate infant feeding practice in Eastern zone of Tigray, Ethiopia: A comparative cross-sectional study”. We have revised and incorporated the changes in the manuscript to address all the insightful suggestions raised by the editors and the reviewers. 

PONE-D-21-02467R3

THE RELATIONSHIP BETWEEN POSTPARTUM DEPRESSION AND APPROPRIATE INFANT FEEDING PRACTICE IN EASTERN ZONE OF TIGRAY, ETHIOPIA: A COMPARATIVE CROSS-SECTIONAL STUDY

PLOS ONE

Dear Dr. Weldu,

Thank you for submitting your manuscript to PLOS ONE. After careful consideration, we feel that it has merit but does not fully meet PLOS ONE’s publication criteria as it currently stands. Therefore, we invite you to submit a revised version of the manuscript that addresses the points raised during the review process.

We look forward to receiving your revised manuscript.

Kind regards,

Melissa F. Young, Ph.D.

Academic Editor

PLOS ONE

Journal Requirements:

Please adjust the reviewers' comments.

Reviewers' comments:

Reviewer's Responses to Questions

Comments to the Author

1. If the authors have adequately addressed your comments raised in a previous round of review and you feel that this manuscript is now acceptable for publication, you may indicate that here to bypass the “Comments to the Author” section, enter your conflict of interest statement in the “Confidential to Editor” section, and submit your "Accept" recommendation.

Reviewer #1: All comments have been addressed

Reviewer #3: (No Response)

2. Is the manuscript technically sound, and do the data support the conclusions?

Reviewer #1: (No Response)

Reviewer #3: Yes

3. Has the statistical analysis been performed appropriately and rigorously?

Reviewer #1: (No Response)

Reviewer #3: I Don't Know

4. Have the authors made all data underlying the findings in their manuscript fully available?

Reviewer #1: (No Response)

Reviewer #3: Yes

5. Is the manuscript presented in an intelligible fashion and written in standard English?

Reviewer #1: (No Response)

Reviewer #3: No

6. Review Comments to the Author

Reviewer #1: (No Response)

Reviewer #3: (No Response)

Editor’s comment

Comment 1: Any changes to the reference list should be mentioned in the rebuttal letter that accompanies your revised manuscript.

Author response: Thank you, our paper has largely improved after providing another revision. We added more references to provide proper context about proper infant feeding practices and removed unnecessary details from the discussion part. The main purpose of including the references is to explain the magnitude of appropriate infant feeding in the study area to the reader and to provide a glimpse into feeding practice in Ethiopia. These are references that have been added to help explain the outcome variables.

21. Shagaro SS, Mulugeta BT, Kale TD. Complementary feeding practices and associated factors among mothers of children aged 6-23 months in Ethiopia: Secondary data analysis of Ethiopian mini demographic and health survey 2019. Arch Public Heal. 2021;79(1):1–12. 

22. Demilew YM, Tafere TE, Abitew DB. Infant and young child feeding practice among mothers with 0 – 24 months old children in Slum areas of Bahir Dar City, Ethiopia. Int Breastfeed J. 2017;12:26:1–9. 

23. Kassa T, Meshesha B, Haji Y, Ebrahim J. Appropriate complementary feeding practices and associated factors among mothers of children age 6-23 months in. BMC Pediatr. 2016;16(131):1-10. 

24. Fanta M, Cherie HA. Magnitude and determinants of appropriate complementary feeding practice among mothers of children age 6–23 months in Western Ethiopia. PLoS ONE 2020;15(12):e0244277. http://dx.doi.org/10.1371/journal.pone.0244277

25. Yonas F, Asnakew M, Wondafrash M, Abdulahi M. Infant and Young Child Feeding Practice Status and Associated Factors among Mothers of under 24-Month-Old Children in Shashemene Woreda, Oromia Region, Ethiopia. Open Access Libr J. 2015;2,1-15. 

26. Mekonnen M, Kinati T, Bekele K, Tesfa B, Hailu D, Jemal K. Infant and young child feeding practice among mothers of children age 6 to 23 months in Debrelibanos district, North Showa zone, Oromia region, Ethiopia. PLoS ONE. 2021;16(9):e0257758.

27. Hassen SL, Temesgen MM, Marefiaw TA, Ayalew BS, Abebe DD, Desalegn SA. Infant and Young Child Feeding Practice Status and Its Determinants in Kalu District, Northeast Ethiopia: Community-Based Cross-Sectional Study. Nutr Diet Suppl. 2021;13:67–81.

Response to the comments

2. Is the manuscript technically sound, and do the data support the conclusions?

Reviewer #1: (No Response)

Author response: Thank you for your suggestion. To ensure reliable information was collected the principal investigator provides proper training for data collectors, and then supervisors check the data accuracy and completeness. In addition, the tools used to measure the outcome were based on WHO core indicators for appropriate infant feeding practice. We have Clearly defined all outcomes (appropriate infant feeding practice) and exposures (postpartum depression). The measurement for infant feeding practice and postpartum depression was contextualized and checked for reliability (internal consistency). Moreover, we have ensured the data comparability between postpartum depression and non-depressed mothers by at least taking participants of control from the same area and checking whether there is a significant difference in sociodemographic factors between postpartum depressed and non-depressed mothers. Thus, the information helped us to achieve the purpose of the study, which is to assess the magnitude of appropriate infant feeding practice and to determine the effect of postpartum depression on infant feeding practice. So, our conclusion is in line with the objective of the study.

3. Has the statistical analysis been performed appropriately and rigorously?

Reviewer #1: (No Response) 

Reviewer #3: I Don't Know

Author response: Thank you for your concern. During analysis, we tried to control potential confounders using multivariable regression. multicollinearity was checked for all the independent variables that are included in the model that had VIF (variance inflation factor) of less than 1.8 and tolerance greater than 0.52. The data also checked for outliers in continuous variables, none of them were significant after being diagnosed by COOKs distance, and the assumptions of binary logistic regression were checked. In addition, interaction among independent variables was assessed during binary logistic regression, and Hosmer and Lemeshow test was also carried out to check model fitness which was not significant (p-value = 0.95). Then, variables with p<=0.25 in the bivariate analysis were included in the final model to consider public health and clinically important variables. Finally, backward logistic regression was used to determine independent predictors of appropriate infant feeding. The sample size was adequate with a high response rate (97%).

4. Have the authors made all data underlying the findings in their manuscript fully available?

Reviewer #1: (No Response)

Author response: all the necessary data has been uploaded as a supporting information file.

5. Is the manuscript presented in an intelligible fashion and written in standard English?

Reviewer #1: (No Response)

Reviewer #3: No

Author response: Thank you, we have made a significant correction after carefully reviewing the manuscript's entire text for typographical and grammatical errors using Grammarly software. In addition, the font size of the table text has been changed to standard. 

6. Review Comments to the Author

Reviewer #1: (No Response)

Reviewer #3: (No Response)

Author response: none of the findings of the manuscript has been attempted to publish in other journals. 

7. PLOS authors have the option to publish the peer review history of their article (what does this mean?). If published, this will include your full peer review and any attached files.

Do you want your identity to be public for this peer review? For information about this choice, including consent withdrawal, please see our Privacy Policy.

Reviewer #1: No

Reviewer #3: No

---

## [Editor Report · Decision Letter 4]

19 Oct 2022

PONE-D-21-02467R4THE RELATIONSHIP BETWEEN POSTPARTUM DEPRESSION AND APPROPRIATE INFANT FEEDING PRACTICE IN EASTERN ZONE OF TIGRAY, ETHIOPIA: A COMPARATIVE CROSS-SECTIONAL STUDYPLOS ONE

Dear Dr. Weldu,

Thank you for submitting your manuscript to PLOS ONE. After careful consideration, we feel that it has merit but does not fully meet PLOS ONE’s publication criteria as it currently stands. Therefore, we invite you to submit a revised version of the manuscript that addresses the points raised during the review process.

ACADEMIC EDITOR:Please include infant age in weeks as a variable in table 1.for your analysis of appropriate infant feeding practices please separate the sample into <6 months and 6 -12 months since the definition of appropriate feeding is different for these age groupsplease add infant age in weeks as a variable in table 3.for S1 Figure please separate out by age <6 months and 6-12 months Please ensure that your decision is justified on PLOS ONE’s publication criteria and not, for example, on novelty or perceived impact.

We look forward to receiving your revised manuscript.

Kind regards,

Tanya Doherty, PhD

Academic Editor

PLOS ONE
---

## [Author Response · Author response to Decision Letter 4]

14 Dec 2022

Response to Academic editor:

Editor’s comment: Please include infant age in weeks as a variable in table 1

Authors response: Thank you for the comment, we have included infant age in months. In the descriptive statistics, (line 209) we have converted the mean age of the infants from days to months and added infant age in table 1. Converting infant age from days to weeks brings some information to be missed while categorizing as less than six months (0-26 weeks) and six and above months (26.1-52.14 weeks), so to overcome this, we used infant age in months as the main objective of this research is not to understand which age specifically associated with appropriate infant feeding.

Moreover, we used infant age in months, to ensure comparability of this study with others as almost all previously conducted similar researches have categorized infant age in months.

Editor’s comment: for your analysis of appropriate infant feeding practices please separate the sample into <6 months and 6 -12 months since the definition of appropriate feeding is different for these age groups.

Authors response: we have done the analysis separately for the two age groups, below 6 month and 6- 11 months. Consequently, two new tables have been added.

Editors comment: please add infant age in weeks as a variable in table 3.

Authors response: Thank you for the comment, we have added infant age to Table 5 (previously labeled as Table 3). This brings some changes to the values and interpretation of the findings, including the AOR of the variables of the final step of backward LR. For instance, the odds of appropriate infant feeding practice were 1.71 times higher among mothers without postpartum depression compared to their counterparts (AOR = 1.71; 95% CI: 1.08-2.69). This statement from the result part was changed to " The odds of appropriate infant feeding practice were 2 times higher among mothers without postpartum depression compared to their counterparts (AOR = 2.03; 95% CI: 1.29-3.18)." due to the addition of infant age as a variable in Table 5.

In addition, the effect of birth order was also changed from “The odds of appropriate infant feeding practices among infants with a birth order above three was 48% (AOR = 0.52; 95% CI: 0.28-0.97) less than those infants with a birth order of three and below” to “The odds of appropriate infant feeding practices among infants with a birth order above three was 58% (AOR = 0.42; 95% CI: 0.26-0.97) less than those infants with a birth order of three and below”.

Corresponding to these changes, in the abstract section, similar changes was made.

Editors comment: for S1 Figure please separate out by age <6 months and 6-12 months

Authors response: Thank you for the comment, we have separately represented appropriate infant feeding among the age groups, 6 months and 6-11 months, using a bar graph.

---

## [Editor Report · Decision Letter 5]

21 Dec 2022

THE RELATIONSHIP BETWEEN POSTPARTUM DEPRESSION AND APPROPRIATE INFANT FEEDING PRACTICE IN EASTERN ZONE OF TIGRAY, ETHIOPIA: A COMPARATIVE CROSS-SECTIONAL STUDY

PONE-D-21-02467R5

Dear Dr. Weldu,

We’re pleased to inform you that your manuscript has been judged scientifically suitable for publication and will be formally accepted for publication once it meets all outstanding technical requirements.

Kind regards,

Tanya Doherty, PhD

Academic Editor

PLOS ONE
---

## [Editor Report · Acceptance letter]

2 Jan 2023

PONE-D-21-02467R5 

The relationship between postpartum depression and appropriate infant feeding practice in Eastern zone of Tigray, Ethiopia: A comparative cross-sectional study 

Dear Dr. Weldu:

I'm pleased to inform you that your manuscript has been deemed suitable for publication in PLOS ONE. Congratulations! Your manuscript is now with our production department. 

Kind regards, 

on behalf of

Professor Tanya Doherty 

Academic Editor

PLOS ONE